# Titanium Dioxide Nanoparticles Exacerbate Allergic Airway Inflammation via TXNIP Upregulation in a Mouse Model of Asthma

**DOI:** 10.3390/ijms22189924

**Published:** 2021-09-14

**Authors:** Je-Oh Lim, Se-Jin Lee, Woong-Il Kim, So-Won Pak, Changjong Moon, In-Sik Shin, Jeong-Doo Heo, Je-Won Ko, Jong-Choon Kim

**Affiliations:** 1College of Veterinary Medicine (BK21 FOUR Program), Chonnam National University, Gwangju 61186, Korea; dvmljo@gmail.com (J.-O.L.); xhdhksdl123@naver.com (S.-J.L.); dvmwoong@gmail.com (W.-I.K.); sowonp0112@gmail.com (S.-W.P.); moonc@chonnam.ac.kr (C.M.); dvmmk79@gmail.com (I.-S.S.); 2Bioenvironmental Science & Technology Division, Korea Institute of Toxicology, Jinju 52834, Korea; jdher@kitox.re.kr; 3College of Veterinary Medicine, Chungnam National University, Daejeon 34131, Korea

**Keywords:** titanium dioxide nanoparticle, asthma, airway inflammation, thioredoxin-interacting protein, apoptosis

## Abstract

Titanium dioxide nanoparticles (TiO_2_NPs) are widely used in industrial and medicinal fields and in various consumer products, and their increasing use has led to an increase in the number of toxicity studies; however, studies investigating the underlying toxicity mechanism have been rare. In this study, we evaluated potential toxic effects of TiO_2_NPs exposure on lungs as well as the development of asthma through the ovalbumin (OVA)-induced mouse model of asthma. Furthermore, we also investigated the associated toxic mechanism. TiO_2_NPs caused pulmonary toxicity by exacerbating the inflammatory response, indicated by an increase in the number and level of inflammatory cells and mediators, respectively. OVA-induced asthma exposed mice to TiO_2_NPs led to significant increases in inflammatory mediators, cytokines, and airway hyperresponsiveness compared with those in non-exposed asthmatic mice. This was also accompanied by increased inflammatory cell infiltration and mucus production in the lung tissues. Additionally, TiO_2_NPs decreased the expression of B-cell lymphoma 2 (Bcl2) and the expressions of thioredoxin-interacting protein (TXNIP), phospho-apoptosis signal-regulating kinase 1, Bcl2-associated X, and cleaved-caspase 3 were escalated in the lungs of asthmatic mice compared with those in non-exposed asthmatic mice. These responses were consistent with in vitro results obtained using human airway epithelial cells. TiO_2_NPs treated cells exhibited an increase in the mRNA and protein expression of interleukin (IL)-1β, IL-6, and tumor necrosis factor-α with an elevation of TXNIP signaling compared to non-treated cells. Moreover, pathophysiological changes induced by TiO_2_NP treatment were significantly decreased by TXNIP knockdown in airway epithelial cells. Overall, TiO_2_NP exposure induced toxicological changes in the respiratory tract and exacerbated the development of asthma via activation of the TXNIP-apoptosis pathway. These results provide insights into the underlying mechanism of TiO_2_NP-mediated respiratory toxicity.

## 1. Introduction

Air pollutants such as yellow and fine dust, have become a critical social issue and are life-threatening to patients with respiratory diseases. Several studies [1,2,3,4] have particularly reported that titanium dioxide nanoparticles (TiO_2_NPs), components of Asian dust and air pollutants, exacerbate respiratory distress [1,2,3,4]. Moreover, TiO_2_NPs induce an intensive inflammatory response by triggering inflammatory cell migration and pro-inflammatory cytokine secretion, consequently contributing to the development and exacerbation of respiratory diseases [5,6]. However, mechanisms associated with the toxic effects of titanium on the respiratory system and its diseases have not been clearly identified.

Of the known respiratory diseases, asthma is an inflammatory disease of the respiratory airways that affects approximately 300 million people globally [7]. It is characterized by excessive inflammation of bronchi and obstruction of the airflow due to increased immune responses, resulting in varying respiratory symptoms, mainly difficulty in breathing, wheezing, coughing, and tightness in the chest [3,8,9]. Based on its main characteristics, asthma has been divided into molecular pathways/clinical presentations or atopic/non-atopic asthma and more recently, into eosinophilic or non-eosinophilic phenotypes based on biological mechanisms [10]. Proinflammatory cytokines are key players in the development and progression of asthma, and they induce elevated immune responses, resulting in characteristic asthmatic responses such as eosinophilia, airway hyperresponsiveness (AHR), airway remodeling, and mucus production [11]. Previous studies have shown that fine dust and air pollutants aggravate asthma; however, the underlying toxicity mechanism is not well established.

Thioredoxin-interacting protein (TXNIP), a critical regulator of pathological responses, is induced by various of stressors including inflammation, metabolic dysfunction, apoptosis, and lung dysfunction [12,13,14]. TXNIP has been expressed in the lungs of experimental animals exposed to lipopolysaccharide and ovalbumin (OVA), which induce increased inflammatory responses via the activation of inflammasomes [15,16,17]. In contrast, TXNIP is involved in the apoptotic response of lung tissues. Elevation of TXNIP expression by various stimuli results in the activation of apoptotic signaling molecules such as mitochondrial apoptosis signal-regulating kinase 1 (ASK1), B-cell lymphoma 2 associated X (Bax), p38 mitogen-activated protein kinase, and cleaved-caspase 3 (Cas3), which eventually trigger the apoptosis of lung tissues [18,19]. The association between TiO_2_NPs and apoptosis under asthmatic conditions has not been clearly established. Therefore, an in-depth study is needed to understand the apoptosis mechanisms triggered via TXNIP in OVA-induced mice and how TiO_2_NPs pathologically exacerbate the development of asthma.

In this study, we investigated the exacerbation of asthma in response to TiO_2_NP exposure in OVA-induced asthmatic mice and explored the underlying mechanisms involving TXNIP and apoptosis.

## 2. Results

### 2.1. Physicochemical Characterization of TiO_2_NPs

The morphology, primary, and hydrodynamic sizes of TiO_2_NPs are shown in Figure 1. The primary and hydrodynamic sizes in phosphate-buffered saline (PBS) were 48.89 ± 15.49 and 238.94 ± 57.94 nm, respectively. The specific surface area of TiO_2_NPs was 40.45 and 39.38 m^2^/g as determined using the Brunauer–Emmett–Teller (BET) and single point methods, respectively (Table 1). The zeta potential of the TiO_2_NPs was −31.01 mV (Figure 1c). Purity of TiO_2_NPs was measured as 21.35% of Ti and 78.65% of O using energy-dispersive X-ray spectroscopy (Figure 1d). TiO_2_NP suspensions did not show detectable endotoxin levels (data not shown). TiO_2_NP concentrations in the lung tissues were determined using inductively coupled plasma mass spectrometry (ICP-MS) (Table 1); the amount of TiO_2_NPs in the TiO_2_NPs-treated groups was markedly increased in a dose-dependent manner compared with that in the vehicle control (VC) group.

### 2.2. Effects of TiO_2_NPs on AHR and Inflammatory Cell Counts

In the pulmonary toxicity study, exposure to TiO_2_NPs significantly increased inflammatory cell counts in the bronchoalveolar lavage fluid (BALF) of mice compared with the BALF of VC mice. In particular, a marked increase in the number of neutrophils and macrophages was observed (Appendix A). BALF of the OVA group exhibited a significant increase in inflammatory cell counts, especially that of eosinophils, compared to that of BALF of the VC group (Figure 2a). OVA-induced mice exposed to TiO_2_NPs exhibited elevated counts of eosinophils, macrophages, and neutrophils compared with the OVA group, and this increase occurred in a dose-dependent manner. As shown in Figure 2b, the mean Penh value was augmented in the OVA group compared with the VC group. Furthermore, Penh values exhibited dose-dependent increases in the OVA + TiO_2_NP mice compared with the OVA mice (Figure 2b).

### 2.3. Effects of TiO_2_NPs on Cytokine Production and OVA-Specific IgE Levels in Serum

To determine whether TiO_2_NPs affect the production of inflammatory cytokines in addition to increasing the number of inflammatory cells, cytokine levels in BALF samples were measured. The pulmonary toxicity study revealed that levels of proinflammatory cytokines, namely tumor necrosis factor (TNF)-α, interleukin (IL)-6, and IL-1β, were significantly augmented in TiO_2_NP-treated groups compared with those in the VC group (Appendix A). Levels of TNF-α, IL-6, and IL-1β in BALF were also significantly greater in the OVA group compared with those in the VC group. Exposure to TiO_2_NPs increased levels of TNF-α, IL-6, and IL-1β compared with those in the OVA group; this increase was dose-dependent (Figure 3a–c). Similar to levels of TNF-α, IL-6, and IL-1β, levels of IL-5 and IL-13 in BALF were also higher in the OVA group. Moreover, levels of IL-5 and IL-13 were increased dose-dependently (Figure 3d,e) and the OVA-specific IgE level in the serum was elevated in the TiO_2_NP-treated groups compared to that in the OVA group (Figure 3f).

### 2.4. Effects of TiO_2_NPs on Airway Inflammation and Mucous Secretion in OVA-Induced Mice

Mice exposed to TiO_2_NPs showed an accumulation of inflammatory cells around the alveoli and bronchi as well as increased mucus production (Appendix A). OVA-induced mice showed a significantly higher degree of airway inflammation than VC mice. When the OVA-induced mice were treated with TiO_2_NPs, airway inflammation worsened in a dose-dependent manner compared to that in OVA-induced mice (Figure 4a). Changes in the mucus production index exhibited a pattern similar to changes in airway inflammation wherein mucus production in the TiO_2_NP-treated groups was markedly escalated in a dose-dependent manner compared with that in the OVA group (Figure 4b). Moreover, thickening of the basement membrane and airway smooth muscle, vasodilatation, increased number of blood vessels, and goblet cell metaplasia were observed in OVA-induced mice. These pathological changes were more severe in TiO_2_NP-treated groups (Figure 4a,b).

### 2.5. Effects of TiO_2_NPs on TXNIP and Apoptotic Protein Expression

Immunohistochemistry (IHC) was used to estimate expression levels of TXNIP and cleaved-Cas3 in lung tissues in response to OVA and TiO_2_NP treatment. Lungs of normal mice exposed to TiO_2_NPs showed a dose-dependent increase in the expression of TXNIP and cleaved-Cas3 compared with those of VC mice. Similarly, exposure to TiO_2_NPs led to a surge in the expressions of TXNIP, phospho-ASK1 (p-ASK1), Bax, and cleaved-Cas3 and a decline in the expression of B-cell lymphoma 2 (Bcl2) compared with that in VC mice (Appendix A). TXNIP expression in the OVA group was greater than that in the VC group and increased in a dose-dependent manner in TiO_2_NP-treated groups than in the OVA group (Figure 5a). Similarly, cleaved-Cas3 expression in the OVA group increased compared to that in the VC group and in the TiO_2_NP-treated groups compared with that in the OVA group (Figure 5b). The effects of TiO_2_NPs on TXNIP activation were determined using immunoblotting. As shown in Figure 6a,b, TXNIP expression in the lungs was increased in the OVA group compared with the VC group as were the expressions of p-ASK1, Bax, and cleaved-Cas3. However, Bcl2 expression decreased in the OVA group compared with the VC group. Compared with the OVA group, exposure to TiO_2_NPs induced a marked increase in the expression of TXNIP, p-ASK1, Bax, and cleaved-Cas3 in a dose-dependent manner while the Bcl2 level in TiO_2_NP-treated groups was lower than that in the OVA group.

### 2.6. Effects of TiO_2_NPs on the Production of Proinflammatory Mediators in NCI-H292 Cells

For the in vitro experiments, TiO_2_NP concentrations for treatment groups were decided based on results of the cell viability assay (Figure 7a). Treatment of the TiO_2_NPs significantly elevated the levels of IL-1β, IL-6, and TNF-α in NCI-H292 cells in a dose-dependent manner compared to the untreated cells (Figure 7b–d).

As shown in Figure 8a–d, real-time reverse-transcription polymerase chain reaction (qRT-PCR) results revealed that expression levels of *IL-1β*, *IL-6*, *IL-8*, and *TNF-α* were significantly increased in a dose-dependent manner in NCI-H292 cells treated with TiO_2_NPs compared with those in the control group.

### 2.7. Effects of TiO_2_NPs on TXNIP and Apoptosis Protein Expression in NCI-H292 Cells

Immunoblotting revealed a dose-dependent rise in TXNIP expression and in levels of p-ASK1, Bax, and cleaved-Cas3 in TiO_2_NP-treated cells compared to those in untreated cells (Figure 9a,b). To determine the role of TXNIP in mediating effects of TiO_2_NPs, we transfected NCI-H292 cells with TXNIP-specific small interfering RNA (siRNA). The control siRNA did not have any effect on the increased expression of p-ASK1, Bax, and cleaved-Cas3 and Bcl2 expression decreased with TiO_2_NP treatment; however, treatment with TXNIP-specific siRNA diminished the expression of Bax and cleaved-Cas3 and increased the expression of Bcl2 in TiO_2_NP-treated cells, restoring them to levels similar to those in the control group (Figure 10a,b).

## 3. Discussion

There has been an increase in the number of patients with underlying respiratory diseases, a vulnerable subpopulation who warrant consideration when evaluating the potential respiratory toxicity of various substances [20]. The aim of this study was to examine the effect of TiO_2_NPs on asthma exacerbation and to elucidate its underlying mechanism. We showed that exposure to TiO_2_NPs aggravated asthma, increased TXNIP expression, and activated apoptosis in lungs of OVA-induced mice. Furthermore, TiO_2_NP treatment of NCI-H292 cells led to an upregulation of apoptotic machinery via upregulation in TXNIP.

In this study, exposure to TiO_2_NPs increased the inflammatory response in the respiratory tract and aggravated major asthma symptoms, namely, airway inflammation, airway remodeling, mucus overproduction, and AHR, in mice with OVA-induced asthma. Eosinophilic inflammatory response, which is characteristic of asthma, is known to be induced by IL-4, IL-5, and IL-13 produced by CD4^+^ T helper type 2 (Th2) cells [21]. Furthermore, proinflammatory cytokines such as TNF-α, IL-6, and IL-1β, function as growth factors for B cells and play an important role in the differentiation of CD4^+^ Th2 cells [22]. These cytokines have been reported to increase the secretion of mucus by stimulating goblet cells of bronchi, resulting in AHR [23,24]. Asthmatic mice exposed to TiO_2_NPs exhibited a characteristic increase in cytokines, further heightening the aforementioned major asthma symptoms. Moreover, in this study, IgE level was elevated upon exposure to TiO_2_NPs in asthmatic mice, and such an increase has been reported to be associated with the overproduction of Th2 type cytokines [25]. Overproduction of IgE and Th2 cytokines is observed in asthmatic patients with the eosinophilic asthma phenotype as well as in asthmatic animals [15,25]. We observed that cytokine production was significantly increased in NCI-H292 cells treated with TiO_2_NPs, which was similar to our in vivo observations. These results were consistent with histological evidence. Exposure to TiO_2_NPs induced inflammatory cell infiltration into lung tissue and mucus production from goblet cells in a dose-dependent manner and aggravated airway inflammatory responses and mucus production in OVA-induced asthmatic animals. Thus, we demonstrated that TiO_2_NPs caused respiratory toxicity and exacerbated asthma, which is consistent with the findings of previous reports [1,6,26].

TXNIP can directly bind to thioredoxin (TRX) and inhibit its function, leading to the activation of an apoptotic signaling pathway. Under normal conditions, TRX inhibits the activation of ASK1 via formation of a complex. However, activated TXNIP induces the dissociation of this TRX-ASK1 complex, resulting in the activation of ASK1 and, consequently, apoptosis [27,28]. The association between nanoparticles and TXNIP has been reported in several studies [15,16]. For example, exposure to silica dioxide nanoparticles has been shown to worsen asthma and increase pulmonary toxicity via upregulation of TXNIP. In this study, exposure to TiO_2_NPs increased TXNIP expression and activated TXNIP downstream signaling in lungs of normal and asthmatic mice. Furthermore, we found that the TiO_2_NP-activated apoptosis was suppressed by downregulation of the TXNIP gene in human airway epithelial cells. In contrast, upregulation of TXNIP promotes apoptosis by increasing the Bax/Bcl2 ratio and cleaved-caspase 3 expression [19]. Overall, exposure of TiO_2_NPs to mice increased the expression of TXNIP in lungs, demonstrating that TXNIP might be involved in the molecular pathogenesis of asthma. This suggests that TXNIP may be responsible for the aggravating effect of TiO_2_NP-induced apoptosis in asthmatic lungs.

In conclusion, exposure to TiO_2_NPs revealed a marked increase in AHR, inflammatory cytokines and responses, and mucus overproduction, which are typical characteristics of asthma, and these characteristic were correlated with those in the apoptotic signaling pathway (Figure 11). Moreover, we elucidated the aggravating effect of TiO_2_NP inhalation in respiratory tracts with asthma, along with the molecular pathogenesis through the regulation of TXNIP. Thus, our present study provides useful information on new target signaling not yet observed in TiO_2_NP-induced asthmatic lung tissues.

## 4. Materials and Methods

### 4.1. Characterization of TiO_2_NPs

TiO_2_NPs were purchased from Sigma-Aldrich (particle size <25 nm, 637254, St. Louis, MO, USA). We quantified the morphology and size of TiO_2_NPs using transmission electron microscopy (JEM-1210, JEOL, Tokyo, Japan) and scanning electron microscopy (Zeiss EVO-MA10; Carl Zeiss Meditec AG, Jena, Germany) at accelerating voltages of 200 kV and 15 kV, respectively. The specific surface area of TiO_2_NPs was measured by nitrogen absorption methods based on the multipoint BET method (ASAP2020; Micromeritics, Norcross, GA, USA). The hydrodynamic size and zeta potential of TiO_2_NPs were determined by ELS-8000 (Otsuka Electronic, Tokyo, Japan). The purity of TiO_2_NPs used in the experiment was determined by energy-dispersive X-ray spectroscopy (Rayny EDX-700, Shimadzu, Kyoto, Japan). The endotoxin levels in thee TiO_2_NP suspension were determined using a Pierce LAL Chromogenic Endotoxin Quantitation Kit (Thermo Fisher Scientific, Waltham, MA, USA). After completion of treatment procedures, lung tissue was harvested, weighed, and digested overnight with concentrated nitric acid, and the resultant samples were analyzed for elemental TiO_2_NPs using ICP-MS (Perkin Elmer, Waltham, MA, USA).

### 4.2. Experimental Procedure for Allergic Asthma Induction

Specific pathogen-free female BALB/c mice (six weeks old) were purchased from Samtako Co. (Osan, Korea) quarantined and acclimatized for seven days. Animals were confined at 22 ± 2 °C in a room with a relative humidity of 50 ± 5%, artificial lighting from 08:00–20:00, and 13–18 air changes per hour. Animals were provided *ad libitum* access to a standard laboratory diet (Samtako Co., Osan, Korea) and water. All experimental procedures were carried out in accordance with the National Institute Health Guidelines for the Care and Use of Laboratory Animals. The Institutional Animal Care and Use Committee of Chonnam National University approved experimental protocols involving animals (CNU IACUC-YB-2020-19).

To investigate pulmonary toxicity of TiO_2_NPs, 24 healthy female mice were randomly assigned to four experimental groups (*n* = 6 per group); VC group and three TiO_2_NPs-treated (5, 10, and 20 mg/kg, respectively) groups. On days 1, 3, and 5, animals of the TiO_2_NP treated groups (5, 10, and 20 mg/kg doses in 50 μL of PBS, respectively) received TiO_2_NPs via intranasal instillation under light anesthesia using Zoletil 50^®^ (Virbac Laboratories, Carros, France). The VC group received 50 μL of PBS via intranasal instillation. TiO_2_NPs were prepared in PBS and sonicated in an ultrasonicator (VCX-130, Sonics and Materials, Newtown, CT, USA) for 3 min (130 W, 20 kHz, pulse 59/1) before intranasal instillation.

To investigate the effect of TiO_2_NPs on the development of asthma, 30 animals were randomly assigned to five experimental groups (each group, *n* = 6); VC group, OVA group, and three OVA+ TiO_2_NPs (5, 10, and 20 mg/kg) groups. On days 1 and 15, mice were sensitized with an intraperitoneal injection of 20 μg of OVA (Sigma-Aldrich, St. Louis, MO, USA) emulsified with 2 mg of aluminum hydroxide (Thermo Scientific, Waltham, MA, USA) in 200 μL of PBS (pH 7.4). On days 22, 24, and 26, mice received a 1 h airway challenge with 1% (*w*/*v*) OVA solution aerosolized using an ultrasonic nebulizer (NE-U12; Omron Corp., Tokyo, Japan). On days 21, 23, and 25, mice of the TiO_2_NP treatment groups (5, 10, and 20 mg/kg doses in 50 μL of PBS, respectively) received TiO_2_NPs via intranasal instillation under light anesthesia using Zoletil 50^®^ (Virbac Laboratories). The VC and OVA group animals received 50 μL of PBS via intranasal instillation. TiO_2_NPs were prepared in PBS and sonicated in an ultrasonicator for 3 min before intranasal instillation.

### 4.3. Measurement of Airway AHR

Penh values were indirectly assessed 24 h after the final intranasal instillation via single-chamber whole body plethysmography (Allmedicus, Seoul, Korea). Mice were anesthetized for a brief period with an intraperitoneal injection of a mixture of Zoletil and Xylazine (40 mg/kg and 10 mg/kg, respectively), subsequently placed in a chamber, and nebulized with aerosolized PBS or methacholine in increasing concentrations (10, 20, and 40 mg/mL).

### 4.4. Collection of Bronchoalveolar Lavage Fluid (BALF) and Cell Counting

Mice were sacrificed at 24 h after measurement of AHR via an intraperitoneal injection of Zoletil 50^®^ (Virbac Laboratories), and a tracheostomy was performed. To obtain BALF, ice-cold PBS (0.7 mL) was infused into their lungs twice and subsequently withdrawn each time using a tracheal cannula (a total volume of 1.4 mL). BALF samples were centrifuged, and its supernatant was collected for biochemical analysis. Collected cells were resuspended in ice-cold PBS (0.5 mL), and 200 μL of the resuspended solution was centrifuged (200× *g*, 4 °C, 10 min) onto slides using a cytospin (Hanil Science Industrial Co., Ltd., Seoul, Korea). Slides were dried, and the cells were fixed and stained. Differential cell counts were performed using the Diff-Quik^®^ staining reagent (Sysmex Corporation, Kobe, Japan) according to the manufacturer’s instructions.

### 4.5. Cytokines Assay

Levels of several cytokines in BALF, namely TNF-α, IL-6, IL-1β, IL-5, and IL-13, were measured using commercial enzyme-linked immunosorbent assay (ELISA) kits (BD Biosciences, San Jose, CA, USA) according to the manufacturer’s protocol. The serum level of OVA-specific IgE was measured using an ELISA kit (BioLegend, San Diego, CA, USA). Absorbance was measured at 450 nm using an ELISA reader (Bio-Rad Laboratories, Hercules, CA, USA).

### 4.6. Histopathology and IHC

After BALF samples were collected, the lung tissue was fixed with 4% (*v*/*v*) paraformaldehyde for 48 h. Tissues were paraffin-embedded, sectioned at a thickness of 4 μm and stained using hematoxylin and eosin (Sigma-Aldrich, St. Louis, MO, USA) or periodic acid-Schiff solution (IMEB Inc., San Marcos, CA, USA) to evaluate airway inflammation and mucus production, respectively. Furthermore, sectioned tissues were processed for IHC, as previously described [29]. The primary antibodies used for the detection of protein expression were anti-TXNIP (NBP1-54578; 1:200 dilution; Novus Biologicals, Littleton, CO, USA) and anti-cleaved-Cas3 (#9661; 1:200 dilution; Cell signaling, Danvers, MA, USA). Each slide was examined manually by investigators blind to the treatment groups using a light microscope (Leica, Wetzlar, Germany) with 10× and 20× objective lenses and a 100× oil immersion lens. Ten randomly selected nonoverlapping areas per slide were captured with a digital camera (IMTcamCCD5; IMT Inc., Daejeon, Korea), and quantitative analyses of airway inflammation, mucus production, and protein expression were performed using an image analyzer (IMT i-Solution software, Vancouver, BC, Canada).

### 4.7. Western Blot Analysis

To quantify protein expression, we performed immunoblotting as previously described [30]. Primary antibodies used are as follows: anti-TXNIP (NBP1-54578; Novus Biologicals), anti-p-ASK1 (SAB4504337; Sigma-Aldrich, St. Louis, MO, USA), anti-total-ASK1 (t-ASK1, ab45178; Abcam, Cambridge, UK), anti-Bcl2 (#2876; Cell Signaling, Danvers, MA, USA), anti-Bax (#2772; Cell Signaling), anti-cleaved-Cas3 (#9661; Cell Signaling), and anti-β-actin (β-act, #4967; Cell Signaling). Densitometric analysis of expression was performed using Chemi-Doc (Bio-Rad Laboratories).

### 4.8. Cell Culture

The human airway epithelial cell line NCI-H292 was obtained from the American Type Culture Collection (Manassas, VA, USA). Cells were grown in RPMI 1640 medium (WELGENE, Gyeongsan, Korea) with 10% fetal bovine serum, streptomycin (100 μg/mL), and penicillin (100 U/mL) and incubated in a humidified chamber maintained at 37 °C with 5% CO_2_. Cells were serum-starved for 1 h before use.

### 4.9. Cell Viability Assay

Cell viability was performed using the EZ-Cytox cell viability assay kit (DAELIL lab, Seoul, Korea), and concentration of TiO_2_NPs was chosen with reference to a previous study [31]. NCI-H292 cells were seeded in 96 well-plates (4 × 10^4^ cells/well). After 24 h, fresh medium and various concentrations of TiO_2_NPs (1.56, 3.13, 6.25, 12.5, 25 µg/mL) were added. The culture plate was incubated for another 24 h. Eventually, viable cells were determined by adding 10 µL of the kit solution to each well before incubating for 4 h. Absorbance was measured at 450 nm using an ELISA reader (Bio-Rad Laboratories, Hercules, CA, USA).

### 4.10. Measurement of mRNA Expression of Proinflammatory Cytokines in NCI–H292 Cells

RNA was isolated using the HiGene Total RNA Prep Kit (Biofact, Daejeon, Korea) according to the manufacturer’s protocol. RNA concentration (A260) and purity (A260/A280 ratio) was measured by spectrophotometry (NanoDrop One Microvolume UV–Vis Spectrophotometer, Thermo Fisher Scientific, Dreieich, Germany), and the ratio for pure RNA A260/280 was higher than 2.0. The total RNA was reverse transcribed into cDNA using a cDNA kit (Qiagen, Hilden, Germany). CFX Connect Real-Time PCR Detection System (1855201, Bio-Rad Laboratories, Hercules, CA, USA) and CFX manager software (1845000 version 3.1, Bio-Rad Laboratories, Hercules, CA, USA) were used to quantify proinflammatory cytokine mRNA expression for RT-qPCR. qRT-PCR experiments were performed using specific forward and reverse primers (Appendix A) and the conditions were as follows: 15 min at 95 °C, 20 s at 95 °C/40 s at 55 °C for 40 cycles, and 10 s at 95 °C/5 s at 65 °C/60 s at 95 °C for the melting curve. The qRT-PCR reaction system was 20 µL: SYBR Premix (Biofact)I, 10 µL; PCR Forward Primer (10 µM), 1 µL; PCR Reverse Primer (10 µM), 1 µL; cDNA template, 2 µL; and distilled water, 6 µL. The mRNA expressions of IL-6, IL-1β, and TNF-α were calculated by the 2^−^^ΔΔCT^ method with the internal reference as GAPDH.

### 4.11. Small Interfering RNA Transfection of NCI-H292 Cells

*TXNIP*-specific siRNA (4392420) and scrambled siRNA (4390843) were purchased from Ambion (Waltham, MA, USA). Each siRNA (20 nM) was transfected into NCI-H292 cells using LipofectamineTM RNAiMAX reagent (Invitrogen, Waltham, MA, USA) following the forward transfection method, as prescribed by the manufacturer. After suppression of endogenous TXNIP expression, cells were treated with 25 µg/mL TiO_2_NPs or PBS and harvested after 6 h. To investigate the protein expression involved in TXNIP-apoptosis signaling, western blot was performed as above-mentioned.

### 4.12. Statistical Analysis

Data were expressed as means ± standard deviation (SD). Statistical significance was determined using analysis of variance followed by Dunnett’s test for multiple comparisons. *p* values less than 0.05 were considered statistically significant.

## Figures and Tables

**Figure 1 ijms-22-09924-f001:**
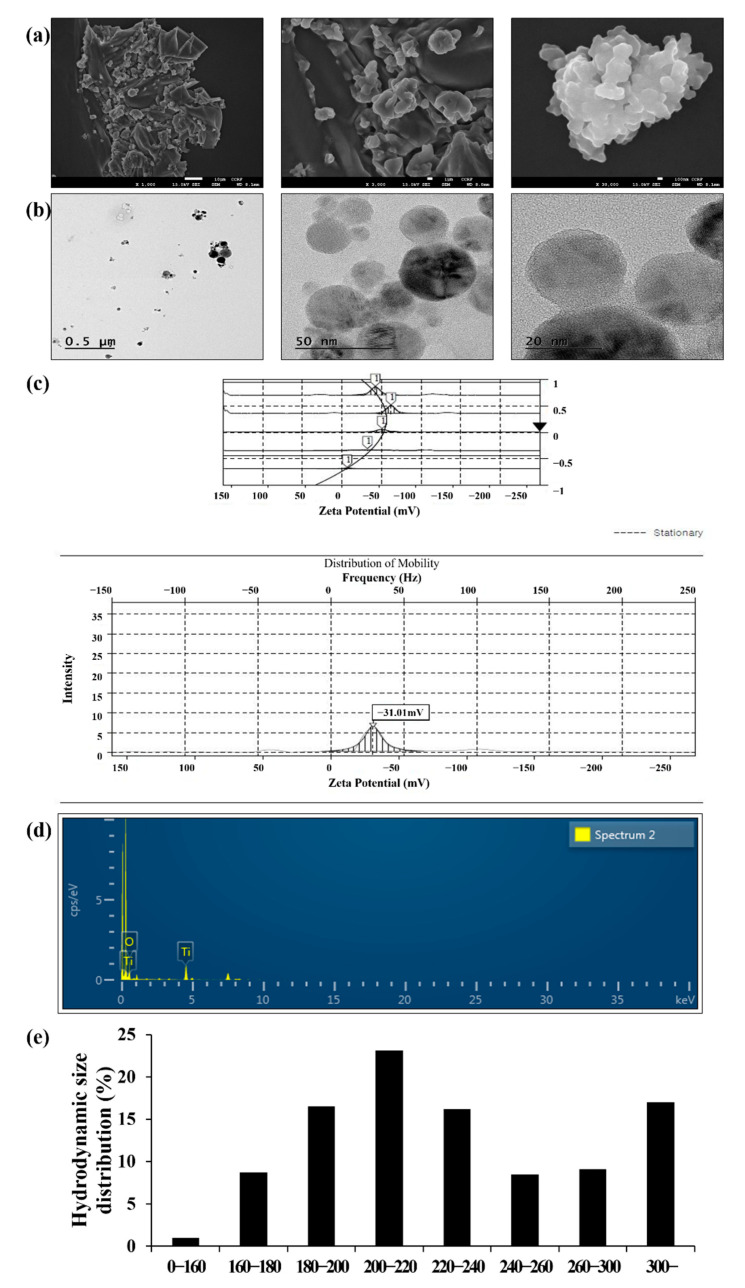
Morphology and physicochemical properties of TiO_2_NPs. (**a**) Morphology of TiO_2_NPs was measured using transmission electron microscopy. Scale bars represent 10, 1, and 0.1 μm, respectively. (**b**) Morphology of TiO_2_NPs was measured using scanning electron microscopy. (**c**) Zeta potential of TiO_2_NPs was measured using ELS-8000 (−31 mV). (**d**) Purity of TiO_2_NPs was measured using energy-dispersive X-ray spectroscopy (Ti: 21.35%, O: 78.65%). (**e**) Hydrodynamic size of TiO_2_NPs in PBS solution was measured using ELS-8000.

**Figure 2 ijms-22-09924-f002:**
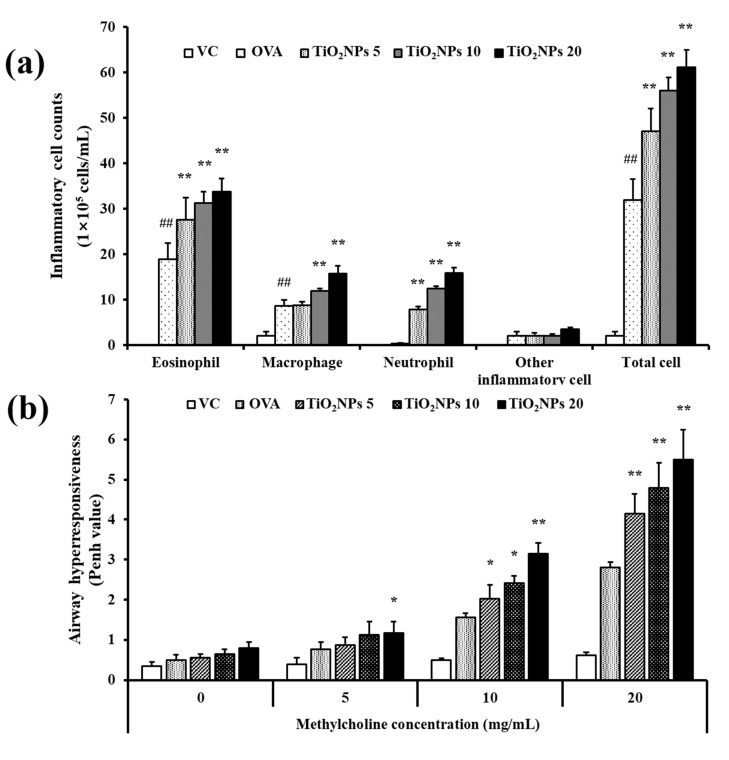
Effects of TiO_2_NP exposure on Penh values and inflammatory cell counts in BALF. (**a**) Penh values. (**b**) Inflammatory cell counts. VC, PBS intranasal instillation; OVA, OVA challenge + PBS intranasal instillation; TiO_2_NPs 5, 10, and 20, OVA challenge +5, 10, and 20 mg/kg of TiO_2_NPs intranasal instillation, respectively. Data are represented as means ± SD, *n* = 6. ^##^ *p* < 0.01, significantly different from the VC group; * *p* < 0.05, ** *p* < 0.01, significantly different from the OVA group.

**Figure 3 ijms-22-09924-f003:**
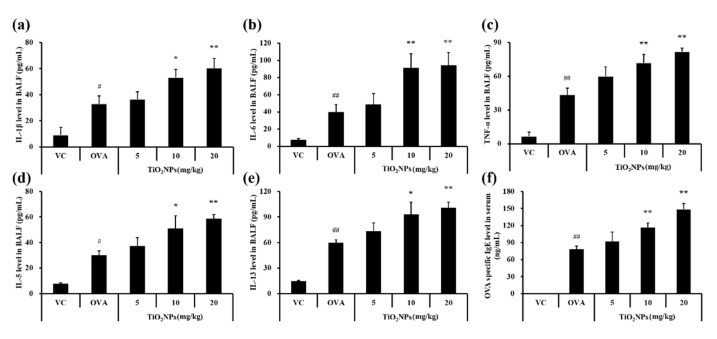
Effects of TiO_2_NP exposure on cytokine levels in BALF and OVA-specific IgE levels in serum. (**a**) IL-1β level in BALF. (**b**) IL-6 level in BALF. (**c**) TNF-α level in BALF. (**d**) IL-5 level in BALF. (**e**) IL-13 level in BALF. (**f**) OVA-specific IgE level in serum. VC, PBS intranasal instillation; OVA, OVA challenge + PBS intranasal instillation; TiO_2_NPs 5, 10, and 20, OVA challenge +5, 10, and 20 mg/kg of TiO_2_NPs intranasal instillation, respectively. Data are represented as means ± SD, *n* = 6. ^#^ *p* < 0.05, ^##^ *p* < 0.01, significantly different from the VC group; * *p* < 0.05, ** *p* < 0.01, significantly different from the OVA group.

**Figure 4 ijms-22-09924-f004:**
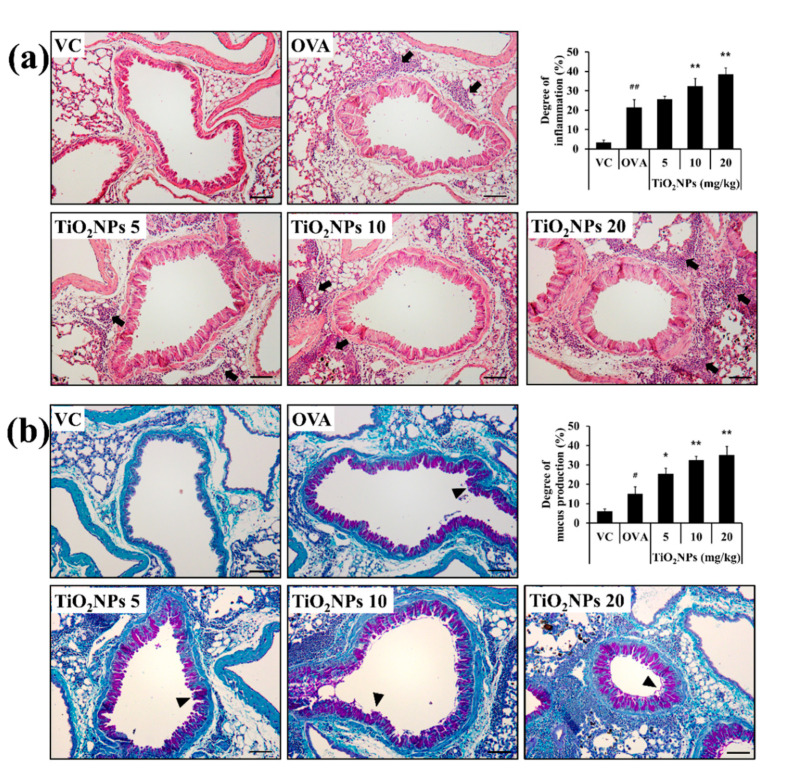
Effects of TiO_2_NP exposure on inflammatory cell infiltration and mucus production in the lungs. (**a**) Lung tissue is stained with hematoxylin and eosin (×200), (**b**) and periodic acid-Schiff stain (×200). VC, PBS intranasal instillation; OVA, OVA challenge + PBS intranasal instillation; TiO_2_NPs 5, 10, and 20, OVA challenge +5, 10, and 20 mg/kg of TiO_2_NPs intranasal instillation, respectively. Black arrows indicate inflammatory cell infiltration. Black arrowheads indicate mucus within lung epithelial goblet cells. Data are represented as means ± SD, *n* = 6. ^#^ *p* < 0.05, ^##^ *p* < 0.01, significantly different from the VC group; * *p* < 0.05, ** *p* < 0.01, significantly different from the OVA group. Bar = 50 μm.

**Figure 5 ijms-22-09924-f005:**
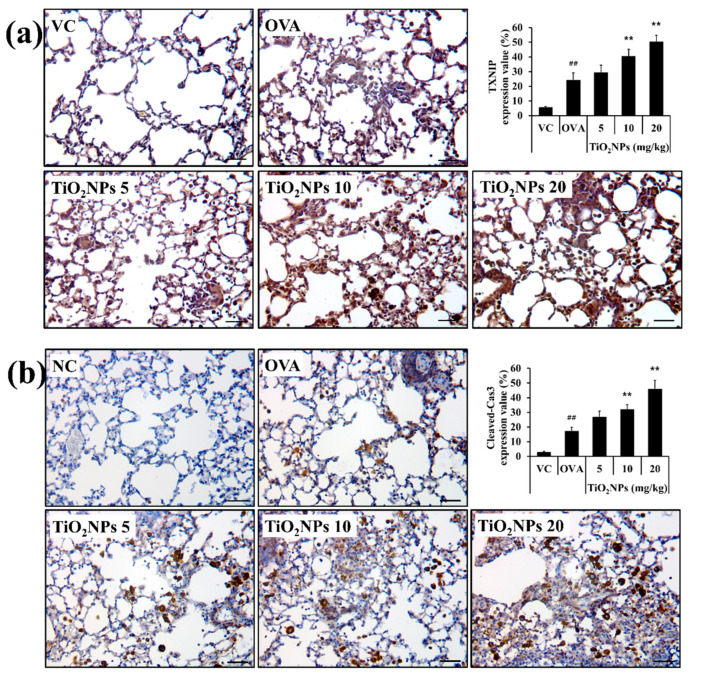
Effects of TiO_2_NP exposure on the expression of TXNIP and cleaved-Cas3 in the lungs. (**a**) Expression of TXNIP (×400, alveolar). (**b**) Expression of cleaved-Cas3 (×400, alveolar). VC, PBS intranasal instillation; OVA, OVA challenge + PBS intranasal instillation; TiO_2_NPs 5, 10, and 20, OVA challenge +5, 10, and 20 mg/kg of TiO_2_NPs intranasal instillation, respectively. Data are represented as means ± SD, *n* = 6. ^##^ *p* < 0.01, significantly different from the VC group; ** *p* < 0.01, significantly different from the OVA group. Bar = 50 μm.

**Figure 6 ijms-22-09924-f006:**
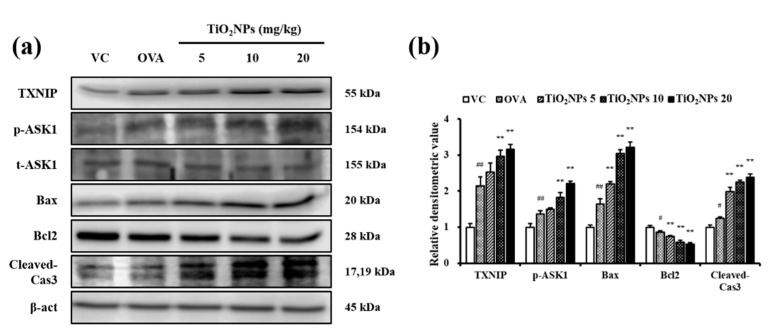
Effects of TiO_2_NP exposure on the expression of TXNIP, p-ASK1, t-ASK1, Bax, Bcl2, and cleaved-Cas3 in the lungs. (**a**) Protein expression was determined using western blotting. (**b**) Relative densitometric values of protein expression. VC, PBS intranasal instillation; OVA, OVA challenge + PBS intranasal instillation; TiO_2_NPs 5, 10, and 20, OVA challenge +5, 10, and 20 mg/kg of TiO_2_NPs intranasal instillation, respectively. Data are represented as means ± SD, *n* = 6. ^#^ *p* < 0.05, ^##^ *p* < 0.01, significantly different from the VC group; ** *p* < 0.01, significantly different from the OVA group.

**Figure 7 ijms-22-09924-f007:**
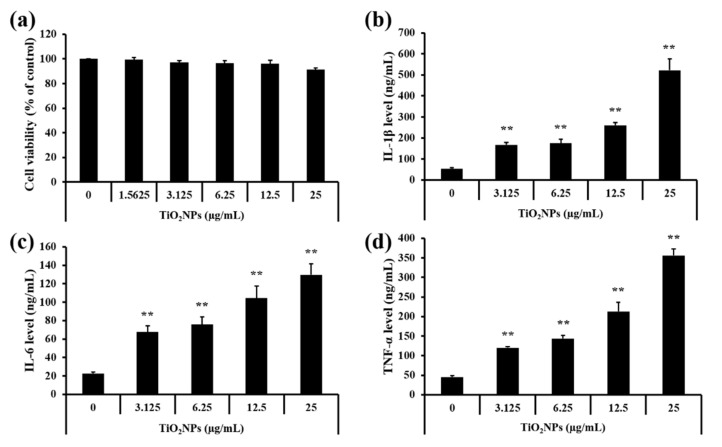
Effects of TiO_2_NP treatment on cell viability and inflammatory cytokines in NCI–H292 cells. (**a**) Cell viability. (**b**) IL-1β level. (**c**) IL-6 level. (**d**) TNF-α level. Control, PBS treatment; 1.5625, 3.125, 6.25, 12.5, and 25 μg/mL of TiO_2_NP treatment; respectively. Data are represented as means ± SD, *n* = 3. ** *p* < 0.01, significantly different from the control group.

**Figure 8 ijms-22-09924-f008:**
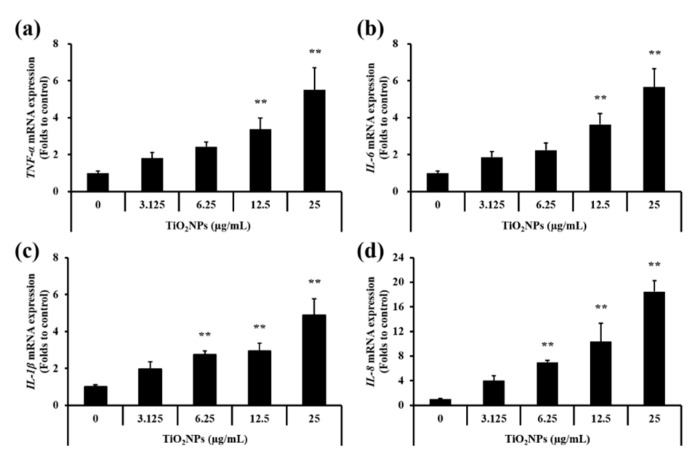
Effects of TiO_2_NP treatment on mRNA expression of inflammatory cytokines measured by qRT-PCR in NCI–H292 cells. (**a**) TNF-α mRNA expression. (**b**) IL-6 mRNA expression. (**c**) IL-1β mRNA expression. (**d**) IL-8 mRNA expression. Control, PBS treatment; 3.125, 6.25, 12.5, and 25 μg/mL of TiO_2_NPs treatment; respectively. Data are represented as means ± SD, *n* = 3. ** *p* < 0.01, significantly different from the control group.

**Figure 9 ijms-22-09924-f009:**
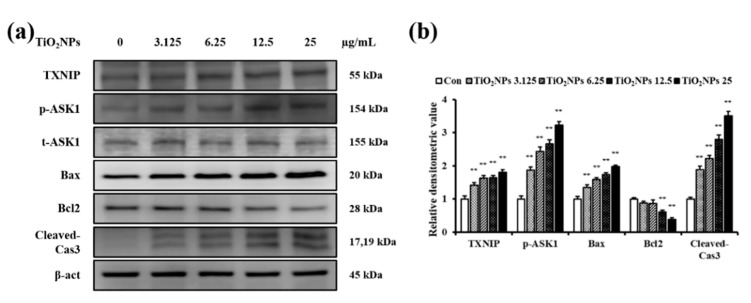
Effects of TiO_2_NP treatment on the expression of TXNIP, p-ASK1, t-ASK1, Bax, Bcl2, and cleaved-Cas3 in NCI–H292 cells. (**a**) Protein expression is determined using western blotting. (**b**) Relative densitometric values of protein expression. Control, PBS treatment; 3.125, 6.25, 12.5, and 25 μg/mL of TiO_2_NPs treatment; respectively. Data are represented as means ± SD, *n* = 3. ** *p* < 0.01, significantly different from the control group.

**Figure 10 ijms-22-09924-f010:**
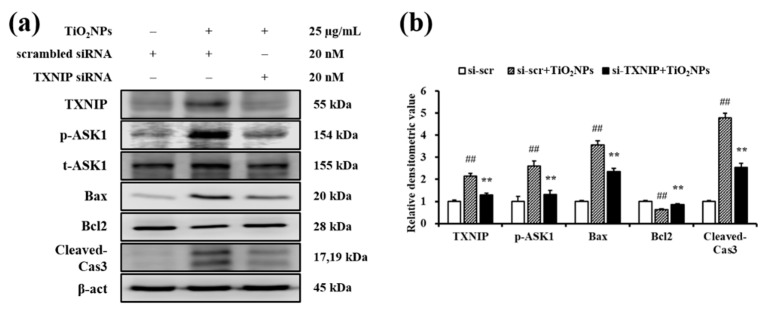
Effects of TXNIP knockdown on TiO_2_NP-induced apoptosis in NCI–H292 cells. (**a**) Proteins expression was determined by western blotting. (**b**) Relative densitometric values of protein expression. Si-scr, scrambled siRNA 20 nM treatment; si-scr + TiO_2_NPs, scrambled siRNA 20 nM + TiO_2_NPs 25 μg/mL treatment; si-TXNIP + TiO_2_NPs, TXNIP siRNA 20 nM + TiO_2_NPs 25 μg/mL treatment. Data are represented as means ± SD, *n* = 3. ^##^ *p* < 0.01, significantly different from the si-scr group; ** *p* < 0.01, significantly different from the si-scr + TiO_2_NPs group.

**Figure 11 ijms-22-09924-f011:**
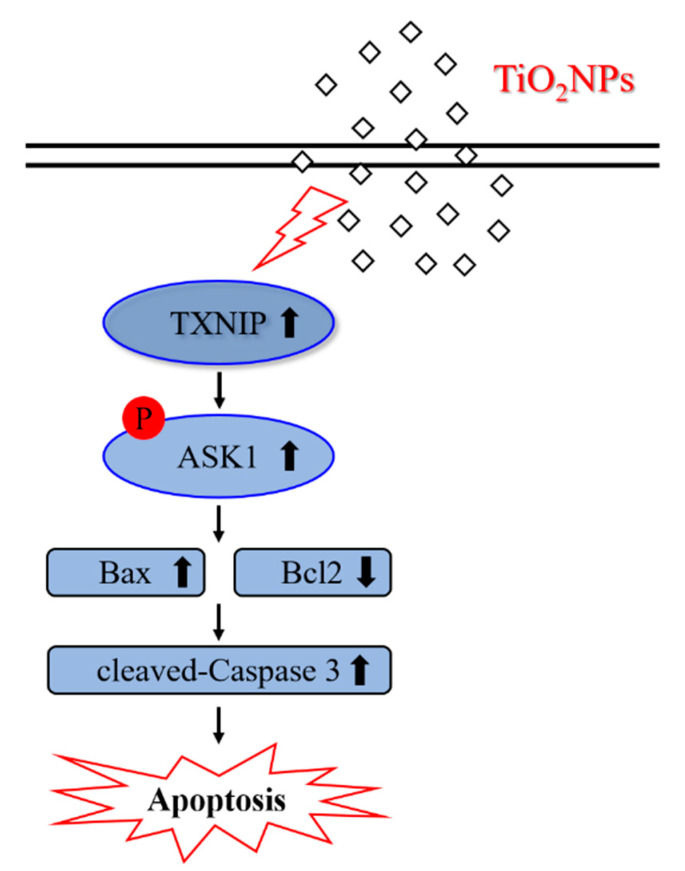
Proposed signaling pathway for TiO_2_NP-induced apoptosis in the lung of asthmatic mice. TXNIP levels are increased by exposure to TiO_2_NP phosphorylates ASK1, thereby increasing apoptosis. TXNIP plays a key mediator role in the exacerbation of asthma by TiO_2_NPs.

**Table 1 ijms-22-09924-t001:** Surface area and ICP-MS measurements of TiO_2_NPs.

BET Surface Area (m^2^/g)	ICP-MS (mg/g)							
40.45	VC		TiO_2_NPs 5		TiO_2_NPs 10		TiO_2_NPs 20	
	0.38 ± 0.032		2.88 ± 0.311		5.68 ± 0.597		6.69 ± 0.613	
**Single point Surface area (m^2^/g)**	VC	OVA		OVA + TiO_2_NPs 5		OVA + TiO_2_NPs 10		OVA + TiO_2_NPs 20
39.38	0.34 ± 0.056	0.32 ± 0.042		2.53 ± 0.397		6.43 ± 0.617		7.94 ± 0.673

Abbreviations: BET, Brunauer–Emmett–Teller; ICP-MS, inductively coupled plasma mass spectrometry; VC, vehicle control group; OVA, ovalbumin challenge + phosphate-buffered saline intranasal instillation group; TiO_2_NP 5, 10, and 20 intranasal instillation groups with 5, 10, and 20 mg/kg of TiO_2_NPs, respectively; and OVA + TiO_2_NPs 5, 10, and 20 intranasal instillation groups of ovalbumin challenge +5, 10, and 20 mg/kg of TiO_2_NPs, respectively.

## Data Availability

Not applicable.

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
