# Peer review of "Titanium Dioxide Nanoparticles Exacerbate Allergic Airway Inflammation via TXNIP Upregulation in a Mouse Model of Asthma"

_ijms, 2021, doi:10.3390/ijms22189924_

Round 1

Reviewer 1 Report

The TiO2 concentration in the animal experiment must be carefully measured and calculated, otherwise, it’s inappropriate to compare the result of other materials. 

For example, the size is at rang of 10- 50 nm in TEM image of  TiO2. The size is over 100 nm in Hydrodynamic size. It need to be addressed.

The element carbon (EC) and carbon black are consider less toxic. Some of low toxic material should be considered as negative control. 

Minor correction

Figure 1

Scale bars should be mentioned in the figure legend in Figure 1 (a) images.

Figure 6 (a)

Other bands in the lane 1 are straight. The shape of lane 1 of TXNIP should be straight as while. (not bending).  

Author Response

  1. The TiO2 concentration in the animal experiment must be carefully measured and calculated, otherwise, it’s inappropriate to compare the result of other materials.
  • We appreciate your comments. We agreed with reviewer’s comment. To determine the concentration of TiO2 in lung tissue, we had performed inductively coupled plasma mass spectrometry (Perkin Elmer, Waltham, MA, USA) for lung tissue. The concentration of TiO2 in lung tissue is appropriate considering injected total amount of TiO2 in this study, which was appropriate compared with previous study (Park et al., 2016)Park et al., 2016. Copper oxide nanoparticles aggravates airway inflammation and mucus production in asthmatic mice via MAPK signaling. Nanotoxicology, 10, 445-452.
  • Reference
  1. For example, the size is at rang of 10- 50 nm in TEM image of TiO2. The size is over 100 nm in Hydrodynamic size. It need to be addressed.
  • As the reviewer comments, the primary size and hydrodynamic size of TiO2NPs are different. In general, nanomaterials have the property of agglomeration when hydrated (Lee et al., 2016). These results can be understood as the intrinsic material properties of nanoparticles. Therefore, the size before and after hydration was presented, and toxicity evaluation was conducted based on these results. To solve this problem, it is necessary to select a non-toxic vehicle that inhibits aggregation during hydration and to conduct animal experiments. This experimental design is expected to lead to other meaningful research results. The evaluation of the toxicity of TiO2NPs according to the type of vehicle is considered to be one of the areas that should be further studied in the future.Lee et al., 2016. Comparative toxicity and biodistribution of copper nanoparticles and cupric ions in rat. Int J Nanomed. 11, 2883-2900.
  • Reference
  1. The element carbon (EC) and carbon black are considered less toxic. Some of low toxic material should be considered as negative control.
  • We appreciate your comments. As commented by reviewer, if the experiment was carried out based on the experimental design in which a group of low toxic materials such as element carbon (EC) and carbon black was added, we considered that more reliable results could be obtained than in this study. In further experiments, we will fully reflect this point.
  1. Figure 1, Scale bars should be mentioned in the figure legend in Figure 1 (a) images.
  • The size of each scale bar is indicated in very small letters on the right side of the scale bar shown in Figure 1(a). Since letter size in picture cannot be change, the scale bar size was additionally listed in the figure legend.
  1. Figure 6 (a), Other bands in the lane 1 are straight. The shape of lane 1 of TXNIP should be straight as while. (not bending).
  • We have improved the figure quality as commented by reviewer.

Reviewer 2 Report

Generally, the results of the study are interesting and the methods used are appropriate, however in order to give consistency to the article additional modifications and clarifications are required:

  1. Asthma is a much more complex syndrome with various pathophysiological mechanisms, or endotypes, driving various clinical presentation forms and specific cytokine storm and in our opinion requires a broader discussion in the introduction section and in the discussion section with reference to the results obtained.
  2. In asthma, there are features of airway inflammation and airway remodeling such as thickening of the basement membrane, smooth muscle hypertrophia and hyperplasia, goblet cell metaplasia and increased angiogenesis. The histological images (lung tissue stained with H&E and lung tissue stained with periodic acid-Schiff, Figure 4 and Figure S3) need to be better explained and discussed in the context of the treatments performed. Also, the results must be correlated with histopathological evaluation of the lungs.
  3. The discussion section does not refer to the results obtained regarding the IgE level in the context of asthma and the performed treatments.
  4. The conclusion is evasive and must be reformulated to support the results.
  5. In the materials and methods section, point 4.10 must be completed with RNA extraction, integrity, purity ratio, PCR and melting curve program, device used, etc. Also, the results must be correlated with histopathological evaluation of the lungs.

Author Response

  1. Asthma is a much more complex syndrome with various pathophysiological mechanisms, or endotypes, driving various clinical presentation forms and specific cytokine storm and in our opinion requires a broader discussion in the introduction section and in the discussion section with reference to the results obtained.
  • We appreciate your comments. As commented by reviewer, we have described in more detail the content related to asthma.
  1. In asthma, there are features of airway inflammation and airway remodeling such as thickening of the basement membrane, smooth muscle hypertrophia and hyperplasia, goblet cell metaplasia and increased angiogenesis. The histological images (lung tissue stained with H&E and lung tissue stained with periodic acid-Schiff, Figure 4 and Figure S3) need to be better explained and discussed in the context of the treatments performed. Also, the results must be correlated with histopathological evaluation of the lungs.
  • We have additionally described the pathological changes observed upon exposure to the TiO2NPs in the results section.
  1. The discussion section does not refer to the results obtained regarding the IgE level in the context of asthma and the performed treatments.
  • As commented by reviewer, we have explained and discussed the results obtained regarding the IgE level in the discussion section.
  1. The conclusion is evasive and must be reformulated to support the results.
  • We have thoroughly reviewed the reviewer’s comment, and the conclusion has been rewritten clearly and concisely.
  1. In the materials and methods section, point 4.10 must be completed with RNA extraction, integrity, purity ratio, PCR and melting curve program, device used, etc. Also, the results must be correlated with histopathological evaluation of the lungs.
  • We appreciate your comments. The qPCR method has been described in detail by reflecting the reviewer’s comment. The association between qPCR results and histological changes in the lungs has been additionally described in the discussion section.

Reviewer 3 Report

The author investigated the exacerbation of asthma in response to TiO2NPs exposure in OVA-induced asthmatic mice and explored the underlying mechanisms involving TXNIP and apoptosis. This manuscript was well written, but it is not enough the relationships of results in the in vivo and in vitro study. Our comments are followings

1) It is not enough the relationships of results in the in vivo and in vitro study. Author must investigate the effect of OVA-TiO2NPs on TXNIP in the TXNIP knockout mouse. The in vivo study using TXNIP inhibitor may also support to clarify the mechanism.

2) A scheme helps to explain the relationship between TXNPI and inflammation in lung. Author should add the Scheme.

3) Table 1: Please mention the explanation of the abbreviation, such as OVA-TiO2NPs 5 et al.

4) Figure 2b: Please modify to bar chart, since the scale in x bar is not correct.

5) Figure 7 and 8: How did you determine the concentration of TiO2NPs? Please discuss.

Author Response

  1. It is not enough the relationships of results in the in vivo and in vitro study. Author must investigate the effect of OVA-TiO2NPs on TXNIP in the TXNIP knockout mouse. The in vivo study using TXNIP inhibitor may also support to clarify the mechanism.
  • We appreciate your comments. As commented by reviewer, if the experiment was carried out based on the experimental design using TXNIP knockout mouse or TXNIP inhibitor, we considered that more reliable results could be obtained than in this study. However, it is hard to perform additional experiments related with KO mouse or inhibitors under our laboratory conditions. In further experiments, we will fully reflect this point.
  1. A scheme helps to explain the relationship between TXNIP and inflammation in lung. Author should add the Scheme.
  • As commented by reviewer, the pathway demonstrated in this study have been explained more clearly by adding a schematic representation.
  1. Table 1: Please mention the explanation of the abbreviation, such as OVA-TiO2NPs 5 et al.
  • We have inserted the explanation of the abbreviation in Table 1.
  1. Figure 2b: Please modify to bar chart, since the scale in x bar is not correct.
  • We have improved the figure quality as commented by reviewer.
  1. Figure 7 and 8: How did you determine the concentration of TiO2NPs? Please discuss.
  • The paper referenced to determine the TiO2NPs concentration has been added in the Materials and Methods section.

Round 2

Reviewer 1 Report

The quality of manuscript has been improved.

Author Response

Thank you for your critical review.

  Thank you for your critical review. 귀하의 비판적인 검토에 감사드립니다.   Thanks for your critical assessment. 귀하의 비판적인 평가에 감사드립니다.   전체 결과를 로드할 수 없음 다시 시도 재시도 중...                 Thank you for your critical reiew.        

Reviewer 2 Report

The manuscript is improved compared to the first form. However, we consider that in the legend of Figure 4 and Figure S3 must be clearly detailed the histopathological changes observed compared to the control and to be indicated where possible by arrows. Also, in the materials and methods section, point 4.10 must be completed with the PCR and the melting curve programs used and in the table S1 must be mentioned the annealing temperature of the primers and the melting temperature of the obtained amplicons.

Author Response

We appreciate your comments. As commented by the reviewer, the histopathological changes observed in the test group are indicated by arrows in the Figure 4 and Figure S3 for easy comparison with the control group. Also, the qPCR method has been described in detail by reflecting the reviewer’s comment, and the annealing temperature and the melting temperature have mentioned in the Table S1. The manuscript was reviewed and revised by professional editors to ensure language and grammar accuracy.

Reviewer 3 Report

The data using NCI-H292 cells don’t reflect the data in ovalbumin (OVA)-induced mouse model of asthma, since NCI-H292 cells was not cultured under the environment similar to asthmatic conditions. Therefore, detail examination (TXNIP knockout or inhibitor) in the OVA-induced mouse model or experiments using cultured cells under asthmatic conditions are required to clarify the effect of TiO2NPs during asthma.

Author Response

We appreciate your comments. We also agreed with reviewer’s comments. As commented by reviewer, environmental condition of in vitro and in vivo was different in this study. However, it was difficult to reflect the concepts of sensitization and challenge applied in vivo to creating an asthmatic environment in an in vitro testing. So, based on previous study (Jung et al. 2012), we have performed this experiment. We have investigated the signaling pathway changed by TiO2NPs using H292 cells and then targeted molecules were applied to OVA-induced asthmatic mouse model.

Reference

Jung et al., 2012. Effect of Asian sand dust on mucin production in NCI-H292 cells and allergic murine model. Otolaryngol Head Neck Surg. 146, 887-894.

Round 3

Reviewer 3 Report

The authors answered all questions.